**Knowledgebase & Database Resources**

# FlyBase: updates to the *Drosophila* genes and genomes database

Arzu Öztürk-Çolak [ID],[1,]* Steven J. Marygold [ID],[1] Giulia Antonazzo,[1] Helen Attrill [ID],[1]
Damien Goutte-Gattat [ID],[1] Victoria K. Jenkins,[2] Beverley B. Matthews,[2] Gillian Millburn,[1] Gilberto dos Santos,[2]
Christopher J. Tabone [ID] [2]; the FlyBase Consortium

[1]Department of Physiology, Development and Neuroscience, University of Cambridge, Cambridge CB2 3DY, UK
[2]Department of Molecular and Cellular Biology, Harvard University, Cambridge, MA 02138, USA

*Corresponding author: Department of Physiology, Development and Neuroscience, University of Cambridge, Downing Street, Cambridge CB2 3DY, UK. Email: ao493@cam.ac.uk

FlyBase (flybase.org) is a model organism database and knowledge base about *Drosophila melanogaster*, commonly known as the fruit fly. Researchers from around the world rely on the genetic, genomic, and functional information available in FlyBase, as well as its tools to view and interrogate these data. In this article, we describe the latest developments and updates to FlyBase. These include the introduction of single-cell RNA sequencing data, improved content and display of functional information, updated orthology pipelines, new chemical reports, and enhancements to our outreach resources.

Keywords: FlyBase; model organism database; Drosophila

## Introduction

FlyBase, the knowledge base for *Drosophila melanogaster*, is one of the most frequently used model organism databases (Bellen *et al.* 2021) with more than 800,000 page views per month. When it was first established 32 years ago, the main purpose of FlyBase was to provide essential information about Drosophila genes and genomes. Since then, FlyBase has been constantly evolving and improving to meet the needs of its users, while still keeping the fundamental knowledge about *D. melanogaster* up to date.

Here, we present newly added features and tools, as well as updates in the areas of expression, function, orthology, and reagents released since our last review (Gramates *et al.* 2022). Aiming to provide ease of access to newly generated expression data, we have added single-cell RNA sequencing (scRNA-seq) and FlyAtlas 2 RNA-seq data. To facilitate the usability of functional data in FlyBase, we have implemented 2 new Gene Ontology (GO) tools—GO summary ribbons and PANGEA—and introduced graphics to illustrate enzymatic reactions. We have also upgraded our orthology data using the latest releases of DRSC Integrative Ortholog Prediction Tool (DIOPT) and OrthoDB. The Variant Effect Predictor (VEP) tool from the European Bioinformatics Institute (EBI) has been integrated in order to provide an alternative view of reported mutations in fly genes. Considering the importance of chemical testing in Drosophila research, we have created Chemical Reports in FlyBase. Last, we have created the Fly Lab List to compile details of all active fly labs around the world, and we have revamped our "New to Flies" resource to provide more help to novice fly researchers.

## Expression

### scRNA-seq data in FlyBase

Since 2021, we have been collaborating with the EMBL-EBI Single-Cell Expression Atlas (SCEA) (Papatheodorou *et al.* 2019)

to curate fly scRNA-seq datasets. As a result of this collaboration, 2 new features have been added to our Gene Report pages to give our users an immediate overview of the cell type–specific gene expression data that the Fly Cell Atlas project provides (Li *et al.* 2022). The first new feature is the "cell type ribbon" (Fig. 1a). Located alongside the preexisting "anatomy ribbon" and "stage ribbon" at the top of the Expression Data section, it shows a selection of about 20 high-level cell types (e.g. neuron, glial cell, muscle cell, epithelial cell, etc.). For each cell type, the corresponding tile is colored depending on the proportion of cells of that cell type in which the Fly Cell Atlas project found the gene to be expressed—the brighter the tile, the more cells of that cell type express the gene. Hovering the mouse cursor over a tile will bring a pop-up frame showing the actual percentages of expressing cells for the various cell subtypes. Under the ribbon, a link allows the user to find on the SCEA website all the scRNA-seq datasets (not only the Fly Cell Atlas dataset) that contain data for the current gene.

The second new feature is the Fly Cell Atlas bar chart (Fig. 1b). Located in the High-Throughput Expression Data subsection, it shows the same selection of high-level cell types as in the cell type ribbon, but instead of the cell proportion, it displays the average level of expression of the gene in those cells that express it. In the High-Throughput Expression Data section, toggle buttons alongside the graph allow the style of the graph and the scale of the bars to be changed. Choosing the "% ↔ level" style option displays a back-to-back plot with both cell proportion and average expression level shown. Possible future iterations of this graph may also allow a selective display of female-only, male-only, or mixed-sex data.

For users who want a deeper look at the expression data from scRNA-seq datasets, but without having to explore raw data directly, we provide "summarized expression data" as a downloadable file in the Genes section of http://flybase.org/downloads/bulkdata,

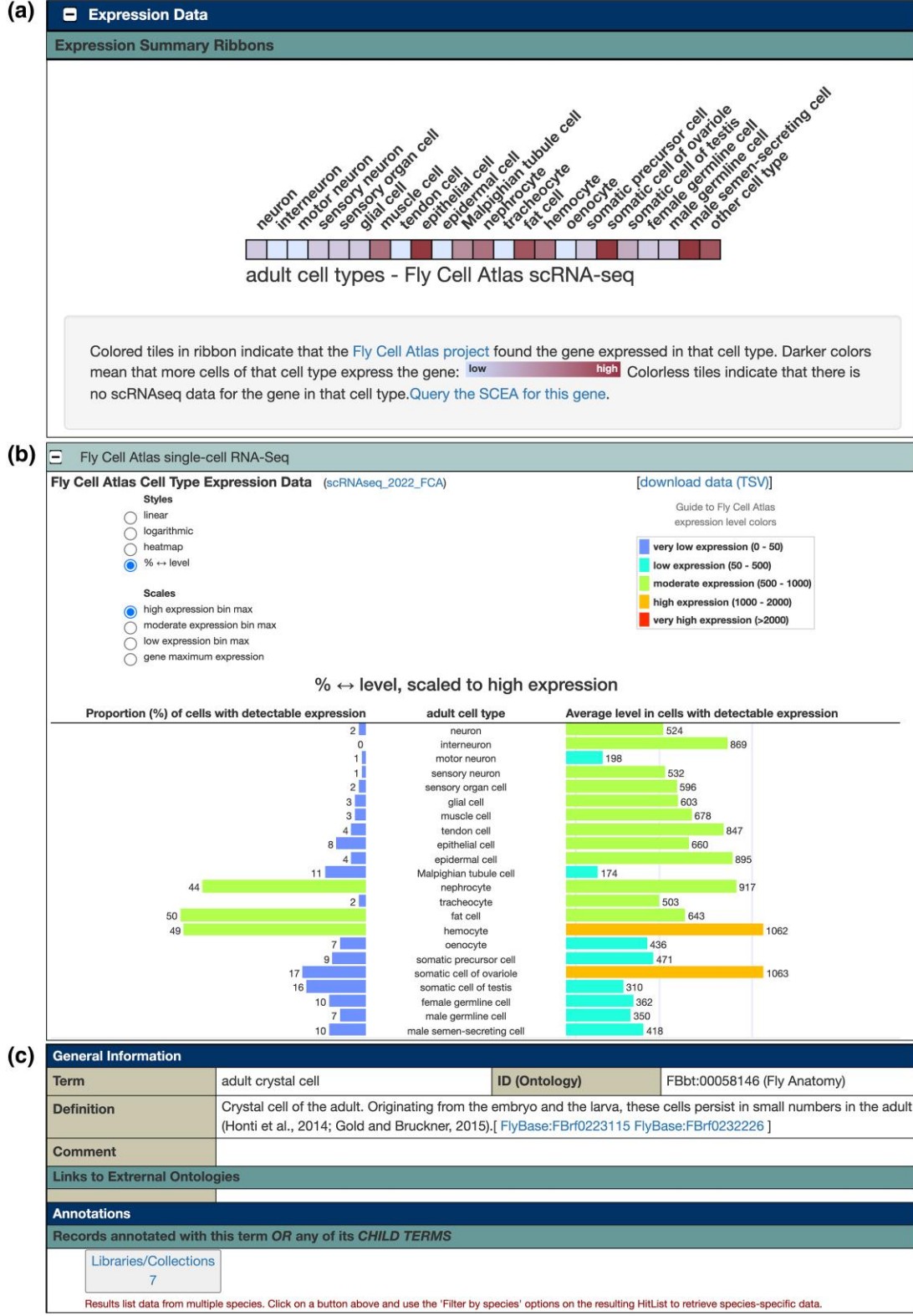

**Fig. 1.** Fly Cell Atlas scRNA-seq data in FlyBase. a) Expression Summary Ribbons subsection of the *srp* Gene Report showing Fly Cell Atlas scRNA-seq cell type ribbon. Each tile is colored depending on the proportion of cells of the indicated cell type in which the current gene is expressed, according to the Fly Cell Atlas dataset. b) High-Throughput Expression Data subsection of the *srp* Gene Report showing Fly Cell Atlas bar chart. For each high-level cell type, this graph displays both the proportion of expressing cells and the average level of expression in all cells that do express the gene. c) The Term Report page for the "adult crystal cell" cell type. Clicking on the "Libraries/Collections" button brings up the list of all scRNA-seq clusters in which adult crystal cells have been identified.

available from the Downloads menu bar button. For each tuple (gene, cell type) in every dataset we have curated, that file contains the same values that would be displayed by the Fly Cell Atlas bar chart: one value for the proportion of expressing cells, and one value for the average expression level.

Last, users interested in a particular cell type can now find all the scRNA-seq datasets relevant for that particular cell type. For that, they would visit the Term Report page for their cell type of interest (by clicking the Vocabularies button on the FlyBase homepage and looking up and selecting their cell type), expand the Annotations section, and select records of types Libraries/ Collections (Fig. 1c); this would bring a list of all the scRNA-seq clusters in which cells of that cell type were identified. From there, it is possible to reach the Dataset Report for the scRNA-seq dataset from which the cluster comes.

### New RNA-seq expression data

We have added 2 specialized RNA-seq data sets to our existing modENCODE RNA-seq expression data. First, the FlyAtlas 2 project characterizes gene expression in different tissues of larvae and adults, both male and female (Leader et al. 2018). This includes 2 distinct datasets: an analysis of gene expression for most genes and a separate analysis of miRNA gene expression. These 2 FlyAtlas2 datasets are displayed as bar graphs in the High-Throughput Expression Data subsection of the Gene Report Expression Data section: see the FlyAtlas2 Anatomy RNA-seq and FlyAtlas2 Anatomy miRNA RNA-seq sections. There are various viewing options, including a "back-to-back" view that allows for easier comparison of expression differences between males and females.

We have also added miRNA gene expression data derived from a meta-analysis of various independent RNA-seq studies, generated by Eric Lai's lab (Mohammed and Lai 2016). The miRNA gene expression is reported as reads per million miRNA reads for various developmental stages, tissues, and cell lines. The data are presented as bar graphs in the High-Throughput Expression Data subsection of the Gene Report's Expression Data section: Anatomy miRNA RNA-seq, Cell Line miRNA RNAseq, and Development miRNA RNA-seq sections.

## Function

### GO: updates to the annotations and displays

FlyBase has been using the GO to capture gene function for over 2 decades. As well as capturing new information, an important aspect of GO curation is to review and update annotations based on current knowledge. With over 130,000 annotations in the database, this large task is often approached by performing targeted reviews. One such review has focused on noncoding RNAs (ncRNAs). The importance of ncRNAs in various cellular processes has become increasingly clear, particularly with regard to gene silencing by regulatory ncRNAs such as miRNAs and lncRNAs (Frías-Lasserre and Villagra 2017). To increase the GO annotation coverage of ncRNAs, we have incorporated computationally derived annotations based on the RNAcentral's mapping of GO terms to Rfam entries (Kalvari et al. 2021), resulting in 1,031 additional annotations. We have reviewed all 262 genes encoding miRNAs and added annotations describing their mode of action and targeted biological process—over the past 2 years, the annotation coverage of miRNAs has increased from 33 to 46%. With the GO consortium, we have also revised the GO to better describe the biogenesis of ncRNAs and gene silencing mediated by ncRNAs resulting in improved annotation coverage/accuracy within FlyBase. For example, new GO terms have been created to annotate genes involved in "primary piRNA processing" or "secondary piRNA processing" (also known as ping-pong amplification), allowing users to find the components of these very distinct processing pathways.

GO Summary ribbons are graphical summaries of the set of GO annotations made to a gene. The ribbon lists a limited set of high-level GO terms to which annotations are mapped using relationships within the ontology. For each ribbon term, a tile is colored to reflect the number of annotations that map to the term. GO Summary Ribbons have been generated for individual genes and displayed in the Function section of Gene Report pages for many years. We have now extended GO Summary Ribbons to Gene Group and Pathway reports to show the GO Summary ribbon for each member gene stacked to from a grid display. From this display, users can compare the GO annotations made to each gene in these sets, allowing them to look for trends and differences and compare the extent of GO annotation. By mousing over or clicking on each cell, users can see the more detailed annotations that are gathered under that term. To further extend this utility, users can compare up to 100 genes that have been assembled into a FlyBase HitList via the GO ribbon stack viewer option on the Export menu (Fig. 2a–c).

Another way in which we have extended the usability of our GO annotations is via a collaboration with the DRSC Functional Genomics Resource to produce PANGEA (https://www.flyrnai. org/tools/pangea/), a gene set enrichment tool for major model organisms and humans (Hu et al. 2023). Genes from a FlyBase HitList can be directly exported to PANGEA and used to populate the input form (Fig. 2d and e). The tool features GO annotation data and cuts of GO annotation data, such as experimentally evidenced only, excluding high-throughput data and GO subsets (slims), plus other gene sets such as complexes, disease models, phenotypes, pathways, and gene groups, which can be used for enrichment analysis, classification, and comparison.

### Rhea graphics added to gene reports

Recently, we enhanced the annotation and display of Drosophila enzyme data within FlyBase (Larkin et al. 2021). This work significantly improved the accuracy and coverage of catalytic GO annotations and added Enzyme Commission (EC) numbers, names, and reaction descriptions to relevant Gene Reports. To further improve access to and utility of these data, we have now added reaction graphics from the Rhea database (Bansal et al. 2022). These graphics are shown within the Catalytic Activity subsection of the Function section of Gene Reports, which has been reorganized to present an integrated view of EC and Rhea data (Fig. 3a). Information about both Rhea and EC is based on cross references to our GO Molecular Function annotations, thereby ensuring that all these data are kept synchronized and up to date. Links to the source databases are also provided where additional information about the reaction and participants may be obtained.

### Protein and ncRNA structures added to gene reports

Information about the structure of gene products can also inform their function. To this end, we have added images of 3D protein structures predicted by AlphaFold (Varadi et al. 2022) and 2D RNA structures predicted by R2DT/RNAcentral (RNAcentral Consortium et al. 2021; Sweeney et al. 2021) to Gene Reports, found

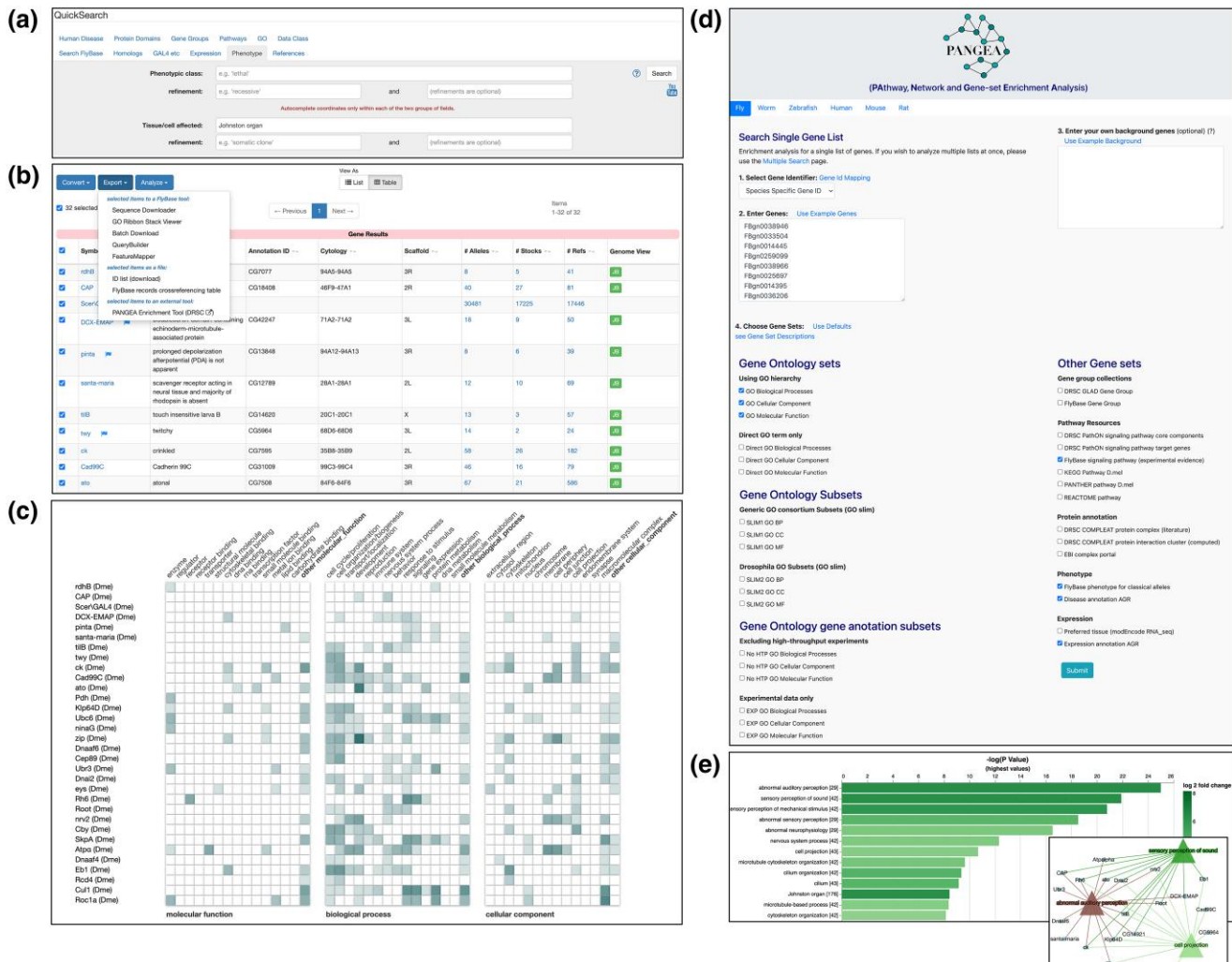

**Fig. 2.** New tools to explore genes from a HitList in FlyBase. a) QuickSearch query for alleles that have a phenotype manifesting in the Johnston organ, gives a HitList of alleles that can be converted into b) a list of genes and exported to c) the GO Ribbon Stack Viewer tool to compare the GO annotations. d) A list of genes exported to the PANGEA tool where e) enrichment over various classifications can be computed and visualized.

within the new Structure subsection of the Gene Model and Products section (Fig. 3b and c). These images can be magnified and repositioned as desired, with resolution down to individual amino acids/nucleotides. Links are also provided to the sources where additional data are available.

## Variant molecular consequences

Reported mutations in fly stocks that can be located on the *D. melanogaster* genome are mapped by FlyBase and displayed in the Nature of the Allele section of the Allele Report under Mutations Mapped to Genome. These include point mutations, deletions, insertions, multinucleotide variants, and delins (sequence alterations, which include an insertion and a deletion, affecting 2 or more bases). We have added an alternative view of these data in the Variant Molecular Consequences section directly underneath, which reports the results of the VEP analysis of FlyBase mutations (Fig. 4; McLaren *et al.* 2016). The analysis is sourced from Ensembl via the Alliance of Genome Resources (Alliance of Genome Resources Consortium *et al.* 2022). Given the genomic coordinates and nucleotide change(s) of a variant, the VEP reports the mutation type and location (using Human Genome Variation Society nomenclature), the consequences to

the gene(s) and affected transcript(s) (e.g. splice_acceptor_variant, stop_gained), and the impact of the change (e.g. high, moderate, modifier). These data are also now displayed under Variant Molecular Consequences in the Variants section of the Gene Report.

## Orthology
### DIOPT v9.1 update

We have updated the local version of the DIOPT (Hu *et al.* 2011) in FlyBase to 9.1. This update includes a compilation of the following orthology prediction algorithms: Compara, eggNOG, Hieranoid, Homologene, InParanoid, OMA, OrthoDB, OrthoFinder, OrthoInspector, OrthoMCL, Panther, Phylome, SonicParanoid, and Domainoid. These algorithms represent a curated subset of the complete collection available from DIOPT and were chosen due to their potential to yield a large volume of orthology calls between *D. melanogaster* and other species of interest at FlyBase. Additionally, DIOPT 9.1 has expanded its species coverage to include new predictions for both *Anopheles gambiae* (African malaria mosquito) and *Escherichia coli* (enterobacterium). Orthology calls for these 2 species are now included in the DIOPT results table in FlyBase. As with previous versions, DIOPT 9.1 predictions can be

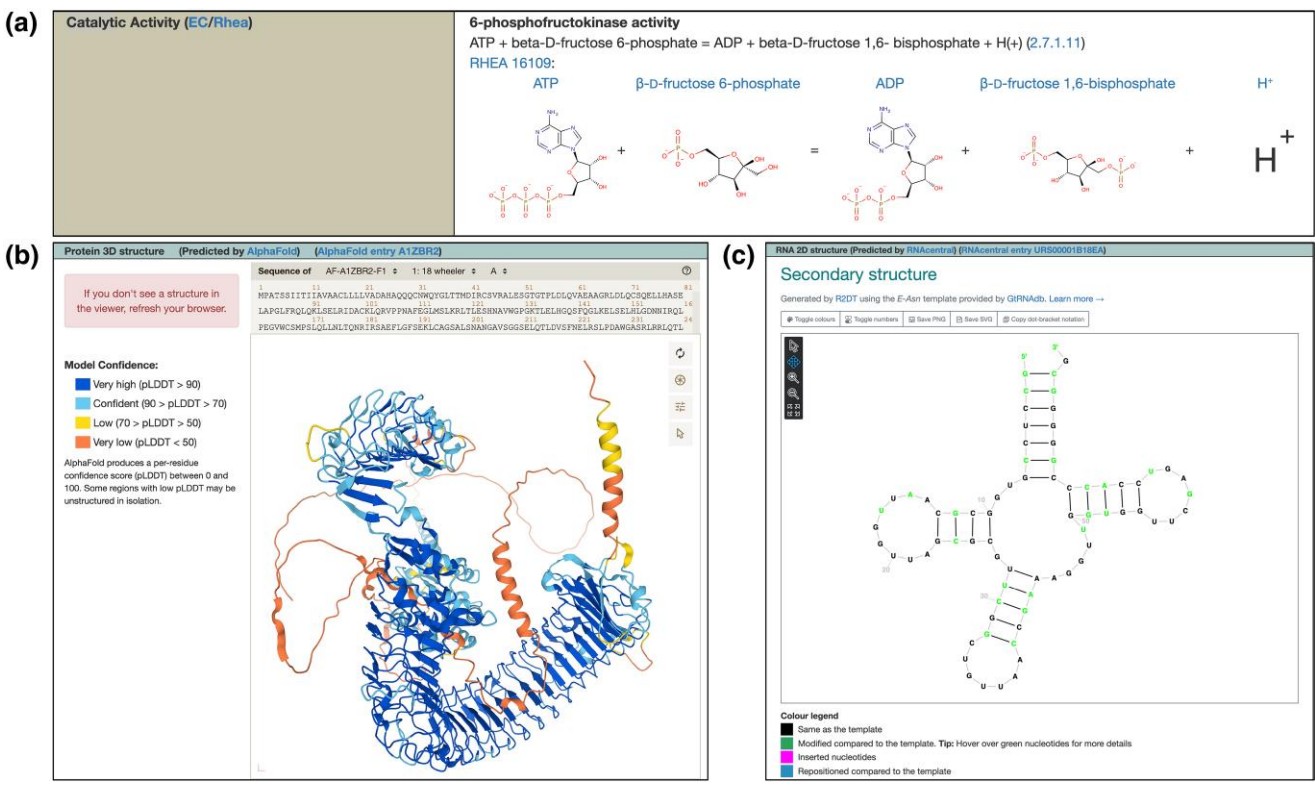

**Fig. 3.** Newly added functional data in FlyBase. a) Catalytic Activity subsection of the *Pfk* Gene Report showing EC and Rhea information. b) Structure subsection of the *18w* Gene Report showing protein structure predicted by AlphaFold. c) Structure subsection of the *tRNA:Asn-GTT-1-9* Gene Report showing ncRNA structure predicted by R2DT.

## Variants

### Variant Molecular Consequences

Variant Effect Predictor (VEP) Analysis of FlyBase variant data. See this glossary for definitions of terms used in the table below. Highlighted text in the table applies to the subject FBid of this report.

| Variant | Affected Genes | Related Alleles | Affected Transcripts |
|---|---|---|---|
| deletion C. > C<br>3R:11,963,124..11,964,369<br>NT_033777.3:g.11963124_11964369del | aurA<br>**consequence:**<br>splice_acceptor_variant,<br>splice_donor_variant,<br>start_lost,<br>5_prime_UTR_variant,<br>intron_variant<br>**impact:** HIGH | aurA<sup>ST</sup> | aurA-RA<br>**consequence:**<br>splice_acceptor_variant,<br>splice_donor_variant,<br>start_lost,<br>5_prime_UTR_variant,<br>intron_variant<br>**impact:** HIGH |
| insertion C > C.<br>3R:11,963,170..11,963,171<br>NT_033777.3:g.11963170_11963171ins | aurA<br>**consequence:**<br>5_prime_UTR_variant<br>**impact:** MODIFIER | aurA<sup>EY03490</sup><br>assoc. with P{EPgy2}aurA[EY03490] | aurA-RA<br>**consequence:**<br>5_prime_UTR_variant<br>**impact:** MODIFIER |
| point_mutation C > T<br>3R:11,963,298<br>NT_033777.3:g.11963298C>T | aurA<br>**consequence:**<br>missense_variant<br>**impact:** MODERATE | aurA<sup>3</sup> | aurA-RA<br>**consequence:**<br>missense_variant<br>**impact:** MODERATE |
| point_mutation A > C<br>3R:11,963,623<br>NT_033777.3:g.11963623A>C | aurA<br>**consequence:**<br>missense_variant<br>**impact:** MODERATE | aurA<sup>3</sup> | aurA-RA<br>**consequence:**<br>missense_variant<br>**impact:** MODERATE |

**Fig. 4.** Variant Molecular Consequences in FlyBase. Variant Molecular Consequences subsection of *aurA* Gene Report.

| General Information | | | |
|---|---|---|---|
| **Name** | bortezomib | **FlyBase ID** | FBch0000415 |
| **ChEBI Name** | bortezomib | **ChEBI ID** | CHEBI:52717 |
| **PubChem Name** | Bortezomib | **PubChem ID** | 387447 (PubChem link: Bortezomib) |
| **Chemical Structure** | | | |
| | bortezomib | | |
| **InChIKey** | GXJABQQUPOEUTA-RDJZCZTQSA-N | | |
| **Definition (ChEBI)** | ChEBI: Bortezomib is l-Phenylalaninamide substituted at the amide nitrogen by a 1-(dihydroxyboranyl)-3-methylbutyl group and at N(alpha) by a pyrazin-2-ylcarbonyl group. It is a dipeptidyl boronic acid that reversibly inhibits the 26S proteasome. It has a role as an antineoplastic agent, a proteasome inhibitor, a protease inhibitor and an antiprotozoal drug. It is an amino acid amide, a member of pyrazines and a L-phenylalanine derivative. It derives from a boronic acid.NCI Thesaurus (NCIt): Bortezomib is a dipeptide boronic acid analogue with antineoplastic activity. Bortezomib reversibly inhibits the 26S proteasome, a large protease complex that degrades ubiquinated proteins. By blocking the targeted proteolysis normally performed by the proteasome, bortezomib disrupts various cell signaling pathways, leading to cell cycle arrest, apoptosis, and inhibition of angiogenesis. Specifically, the agent inhibits nuclear factor (NF)-kappaB, a protein that is constitutively activated in some cancers, thereby interfering with NF-kappaB-mediated cell survival, tumor growth, and angiogenesis. In vivo, bortezomib delays tumor growth and enhances the cytotoxic effects of radiation and chemotherapy. | | |
| **Roles Classification (ChEBI)** | **antineoplastic agent** A substance that inhibits or prevents the proliferation of neoplasms. **antiprotozoal drug** Any antimicrobial drug which is used to treat or prevent protozoal infections. **protease inhibitor** A compound which inhibits or antagonizes the biosynthesis or actions of proteases (endopeptidases). **proteasome inhibitor** A drug that blocks the action of proteasomes, cellular complexes that break down proteins. | | |

**Fig. 5.** Chemical Reports in FlyBase. Bortezomib Chemical Report.

accessed from the FlyBase homepage using the Homologs search tool or viewed on Gene Reports via the Orthologs and Paralogs sections.

## Updates to OrthoDB implementation

*D. melanogaster* orthologs from additional metazoa are shown in the Other Organism Orthologs subsection of Gene Reports. These are sourced from OrthoDB (Kuznetsov *et al.* 2023) and complement the DIOPT data by listing orthologs from other organisms of interest to fly researchers, including other Drosophila species and related insects. Following a survey of our FlyBase Community Advisory Group, OrthoDB-derived orthologs are now retrieved via a live API call rather than being directly integrated within the FlyBase database. One advantage of this

approach is that these data will always reflect the latest OrthoDB release, currently version 11. However, the new implementation also means that OrthoDB data have been removed from JBrowse, precomputed files, GFF files, and QuickSearch aspects of FlyBase.

## Reagents
### Regular updates of stocks reports

We continue to add reports for new genetic reagents generated by a number of large-scale stock providers, to highlight the availability of these reagents to the community and enable linking to the relevant Stock Center. For the Gene Disruption Project (CRIMIC-RMCE insertions, mostly encoding GAL4; Kanca *et al.* 2022), the TRiP-CRISPR

Project (sgRNA transgenes for gene knockout or overexpression; Zirin *et al.* 2020), the Drosophila Humanization Project (UAS human cDNA transgenes; Marcogliese *et al.* 2022), and since 2022, the Fourth Chromosome Resource Project (Stinchfield *et al.* 2023), we create reports as these reagents are submitted to the Bloomington Drosophila Stock Center.

We have also created reports for 2 large collections available from the Korea Drosophila Resource Center: the GenExel collection (a set of mapped P{EP} insertion lines) and the K-Gut Project (GAL4 drivers characterized as being expressed in the gut). Reports have also been prepared for 2 sets of stocks from the National Institute of Genetics Fly Stocks (NIG-Fly): the sgRNA collection (designed to generate gene knockouts) and the knockout (KO) collection of gene knockouts that were generated using the NIG-Fly sgRNA collection.

### Transgenic product class

Transgenic alleles now have a "Transgenic product class" field that describes the nature of the encoded gene product using appropriate Sequence Ontology term(s) (Eilbeck *et al.* 2005), to allow researchers to easily identify different types of transgene for a gene of interest. These terms are used to indicate whether transgenes encode a wild-type product or a sequence targeting reagent (e.g. sgRNA, RNAi reagent) and to describe whether there are structural changes (e.g. missense_variant) and/or functional changes (e.g. dominant_negative_variant) to the encoded gene product.

## Chemicals

The newest data class and associated report in FlyBase is for chemicals. More specifically, we are capturing chemicals used to differentiate experimental conditions, rather than all chemicals used in a reference. For example, this includes chemicals tested for their therapeutic or toxicological properties but does not include chemicals used only as tools to drive a genetic construct or generate new alleles. As FlyBase is not primarily concerned with chemicals, we are incorporating information from 2 well-established chemical databases with different strengths: PubChem (Kim *et al.* 2023), which has a wider range of chemicals and more links to external databases, and ChEBI (Hastings *et al.* 2016), which organizes its entries in multiple ontologies, including those describing biological roles and applications.

Like other FlyBase reports, the Chemical Report is organized into sections. The Chemical Structure, Definitions, and Roles Classification sections provide general information (Fig. 5). Each term in the Roles Classification subsection links to a Term Report, where you can then find other chemicals associated with this term or browse the ontology to find similar terms. The Synonyms section includes alternate names for each chemical found in either the chemical database or in FlyBase papers. All references using the chemical are documented in the References section.

Chemical data at FlyBase are continually under development, both in terms of increasing the number of references curated and in terms of updating the information on the Chemical Report. Our initial wave of curation is focusing on papers with existing disease model curation plus studies of toxicology and olfaction. A future function of this data class will be to include causative or therapeutic chemicals within disease model annotations.

## Outreach resources

### Frequently Asked Questions update

Our Frequently Asked Questions (FAQ) page provides answers to common queries about FlyBase. We have recently reorganized this page to group questions by subject area and updated the content to reflect queries sent to our Help Mail account in the last few years. We have also added a FAQ link to the "Help" menu of the navigation bar and the footer present on every FlyBase webpage.

### "New to Flies" resources

To aid researchers who are new to Drosophila, we have added a "New to Flies" button near the top of the FlyBase homepage, which links to our updated guide to relevant resources at FlyBase and other sites.

### Fly Lab List

We have compiled a Fly Lab List that aims to include all active labs undertaking a substantial fraction of their research using *Drosophila* (any species of *Drosophila*). Our goal is to assist our colleagues to find fly labs in particular geographical locations and to gain an accurate number of fly labs worldwide. Each entry in the list includes the name of the lab head, the lab's location, and a link to the lab website. Over 1,900 entries are included as of mid-2023. The list can be accessed from the News and Outreach section of the External Resource sidebar on the FlyBase homepage and from the "Community" menu of the navigation bar present on every FlyBase webpage. We encourage labs to check whether they are included and represented correctly, and if not, add themselves or revise their entry using the links at the top of the Fly Lab List page.

## Data availability

All FlyBase data and tools are freely available at https://flybase.org/. The DRSC PANGEA tool is freely available at https://www.flyrnai.org/tools/pangea/. Please note that all FlyBase data and images shown in this work are from FB2023_05 release.

## Acknowledgments

We would like to thank the PIs, curators, and developers of FlyBase for their comments on the manuscript. At the time of writing, the members of the FlyBase Consortium included the following: Norbert Perrimon, Susan Russo Gelbart, Kris Broll, Madeline Crosby, Gilberto dos Santos, Kathleen Falls, L. Sian Gramates, Victoria K. Jenkins, Ian Longden, Beverley B. Matthews, Jolene Seme, Christopher J. Tabone, Pinglei Zhou, Mark Zytkovicz, Nick Brown, Giulia Antonazzo, Helen Attrill, Damien Goutte-Gattat, Aoife Larkin, Steven Marygold, Alex McLachlan, Gillian Millburn, Clare Pilgrim, Arzu Öztürk-Çolak, Thomas Kaufman, Brian Calvi, Seth Campbell, Josh Goodman, Victor Strelets, Jim Thurmond, Richard Cripps, and TyAnna Lovato.

## Funding

National Human Genome Research Institute (NHGRI) at the National Institutes of Health (NIH), United States (U41HG000739 and U24HG010859), Medical Research Council, United Kingdom (MR/W024233/1), National Science Foundation, United States (2039324), Wellcome Trust, United Kingdom (PLM13398), Biotechnology and Biological Sciences Research Council, United Kingdom (BBSRC; BB/T014008/1), and Drosophila researchers around the world who support FlyBase through their website access fees.

## Conflicts of interest

The author(s) declare no conflicts of interest.

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

*Editor: V. Wood*