## [Peer Review File · Genetics]

FlyBase: updates to the Drosophila Genes and Genomes database

Arzu Öztürk-Çolak, Steven Marygold, Giulia Antonazzo, Helen Attrill, Damien Goutte-Gattat, Victoria Jenkins, Beverley Matthews, Gillian Millburn, Gilberto dos Santos, and Christopher Tabone

NOTE: The reviews and decision letters are unedited and appear as submitted by the reviewers.

In extremely rare instances and as determined by a Senior Editor or the EIC, portions of a review may be redacted. If a review is signed, the reviewer has agreed to no longer remain anonymous.

The review history appears in chronological order.

Review Timeline:

Submission Date:	2023-10-17
Editorial Decision:	2023-11-06
Revision Received:	2023-11-27
Accepted:	2023-11-27

November 6, 2023

RE: GENETICS-2023-306553

Dear Dr. Öztürk-Çolak:

I am pleased to accept your manuscript entitled "FlyBase: updates to the Drosophila Genes and Genomes database" for publication in GENETICS, pending minor revision.

Please submit your revision along with a response to the reviewers' concerns and suggestions, which can be viewed at the bottom of this email. I expect this can be done within 30 days.

Upon resubmission, please include:

1. A clean version of your manuscript;
2. A marked version of your manuscript in which you highlight significant revisions carried out in response to the major points raised by the editor/reviewers (track changes is acceptable if preferred);
3. A detailed response to the editor's/reviewers' comments and to the concerns listed above. Please reference line numbers in this response to aid the editors.

Additionally, please ensure that your revision is formatted for GENETICS: <https://academic.oup.com/genetics/pages/general-instructions>.

Follow this link to submit the revised manuscript: Link Not Available

Thank you for submitting your research to Genetics.

Sincerely,

Valerie Wood
Associate Editor
GENETICS

Approved by:
Paul Sternberg
Senior Editor
GENETICS

Reviewer comments:

Reviewer #1 (Comments for the Authors (Required)):

Flybase continues to be a gem for the Drosophila research community. This central resource is constantly updated with new findings and with thoughtful and helpful additions of new analytic methods and resources. The latest updates are excellent and important. They are well-described and explained in the present manuscript, and are important to publish both for fly researchers and so databases for other organisms are aware of Flybase's latest updates. While regular users of Flybase have likely already encountered at least some of these updates, it is important to publish this compendium so that researchers who have not discovered these additions know about them.

I have no concerns about this report, though I suggest some minor edits. Only the first one, about the title, is important.

Title: Please include "2023" in the title, since there have been, and will be, other updates.

Line 43: define EBI?

Line 44: I'd say "strains" instead of "stocks".

Line 45: maybe reword as: "...considering the importance of chemical testing in Drosophila research...".

Line 51 would read more smoothly as: "Since 2021 we have been collaborating with the EMBL..."

Line 47: "New to Flies" should be in quotes.

In a few places a ref or two might be helpful, for ex. on line 131.

Line 304: Please call it: "Outreach resources". Otherwise, one expects to hear how Flybase did outreach, which is not what is meant here.

It might be nice to include brief mention of connections to other model-organism databases, though I realize this is not an update per se. Similarly to human-disease genes (ditto).

Reviewer #2 (Comments for the Authors (Required)):

In this knowledge base update paper, the authors describe recent tool innovations, features and data additions to FlyBase. The contents of this manuscript are likely of general interest to a large proportion of fly researchers, and additionally the new developments will be of interest to people working at other knowledge bases.

I am only suggesting a few minor clarifications to the manuscript, and have a few suggestions for improvement. Please see the attached word document.

Associate Editor comments:

It's an informative and well-written update; the recommendations are very minor so I will leave it to your judgment which are included as improvements.

In this knowledge base update paper, the authors describe recent tool innovations, features and data additions to FlyBase. The contents of this manuscript are likely of general interest to a large proportion of fly researchers, and additionally the new developments will be of interest to people working at other knowledge bases.

I am only suggesting a few minor clarifications to the manuscript, and have a few suggestions for improvement.

Section: Single cell RNA sequencing data in FlyBase

I find the font in the ribbon quite difficult to read, it appears almost blurred, and I would prefer a different font. I also find that the diagonal arrangement of the text makes the ribbon difficult to read and link to the tiles. I tried zooming in on the page which makes the font easier to read and I noticed that there is quite a bit of white space to the sides of the ribbon compared to the text box underneath, especially on the left-hand side - perhaps the ribbon could be made a little bit bigger relative to the grey text box underneath?

Personally, and as a suggestion, I would prefer a simple row-by-row horizontal arrangement of tiles and text - this might take up more vertical space, but that shouldn't be a problem since users can navigate the page using the page links menu on the right-hand side.

A horizontal arrangement would also make it easier to compare the manually curated anatomy data to the single cell data. Shading could be used to assist linking anatomy to cell type, eg see sketch below the screenshot.

Colored tiles in ribbon indicate that the Fly Cell Atlas project found the gene expressed in that cell type. Darker colors mean that more cells of that cell type express the gene:
 low high
 Colorless tiles indicate that there is no scRNAseq data for the gene in that cell type. Query the SCEA for this gene.

Colored tiles in ribbon indicate that expression data (RNA and/or protein) has been curated by FlyBase for that anatomical location. Colorless tiles indicate that there is no curated data for that location.

Colored tiles in the ribbon indicate the average RNA expression level of the gene at the indicated stages:
 low high
 as determined by RNA-seq (RPKM) using whole organism samples modENCODE, Brown et al., 2014. For complete stage-specific expression data, view the modENCODE Development RNA-Seq section under High-Throughput Expression below.

■ Neuron	■ Brain
■ Interneuron	
■ Motor cell	
■ Fat cell	■ Adipose system
■ Female cell	■ Reproductive system
■ Male cell	

For the section Fly Cell Atlas single-cell RNA-Seq, I think it would look cleaner if “Styles” “Scales” and “Guide to Fly Cell Atlas expression level colors” were arranged side by side. There’s a lot of white space currently.

Fly Cell Atlas Cell Type Expression Data (scRNAseq_2022_FCA) [download data (TSV)]

Styles

linear

logarithmic

heatmap

% ↔ level

Scales

high expression bin max

moderate expression bin max

low expression bin max

gene maximum expression

Guide to Fly Cell Atlas expression level colors

- very low expression (0 - 50)
- low expression (50 - 500)
- moderate expression (500 - 1000)
- high expression (1000 - 2000)
- very high expression (>2000)

logarithmic, scaled to high expression

cell type	very low expression	low expression	moderate e	high expre
neuron				524

Line 88: Please indicate that the Vocabularies button is located on the main/home/landing page

It felt a bit unintuitive/unexpected that the vocabularies section even existed. It would be user-friendly and more intuitive to find if the cell-type terms were directly linked to their respective term report pages by a mouse-click (i.e., a discreet hyperlink from the gene report page)

Another thing that I found a bit confusing was that when narrowing down my cell-type search to “adult pericerebral fat cell” (6 records) I get the message “Results list data from multiple species. Click on a button above and use the 'Filter by species' options on the resulting HitList to retrieve species-specific data.” But all my results are for *D. melanogaster*? (see second screen-shot). I did not understand that message since it didn’t seem applicable.

Additionally, my personal preference would be to see the results on the subtree results page in a Table rather than the default “list” (list sounds like a table? I think I would call what you are displaying cards). Just my opinion as I strongly dislike card views!

Annotations
 Records annotated with this term OR any of its CHILD TERMS

Libraries/Collections
6

Results list data from multiple species. Click on a button above and use the 'Filter by species' options on the resulting HitList to retrieve species-specific data.

Full annotation statements including this term (annotations to child terms are NOT included), and relevant relationships
 Spanning Tree (Parents/Children) Only view relationship: all Search All Vocabularies

```

adult head
|  _adult pericerebral fat mass
adult fat body
|  _adult pericerebral fat mass
|  _adult fat cell
portion of tissue
|  _adult pericerebral fat mass
fat cell
|  _adult fat cell
adult pericerebral fat cell 6 rec.
  
```

Filter by species clear

- D. melanogaster* (6)
- H. sapiens* (transgenes in flies) (0)
- other *Drosophila* species (0)
- Other species (0)

Filter by data class clear

6 selected New Hitlist

- scRNAseq_2022_FCA**
D. melanogaster
 Class: result
 Title: Clustering a
- scRNAseq_2022_FCA**
D. melanogaster
 Class: result

Section: New RNA-Seq Expression data

Again, there is a lot of white space and perhaps "Styles" "Scales" and the legend could be arranged on the same horizontal space.

It was not intuitive to me that "back to back" would arrange male and female side by side – could the wording be clearer (maybe it is clear to fly people)? Or could you include a tool-tip if the mouse is hovered over the option?

Section: Function

Line 146-153: could you please elaborate a bit more in this section & provide higher resolution screenshots?

I assume that fig 2A relates to finding a set of genes with shared function, since it depicts a phenotype search? I searched for the same thing as shown – e.g. affected tissue: Johnston organ

I got 41 results – 38 from *D. melanogaster* – but your search results show ‘32’ (I know this will change with time and curation, but confusing since your manuscript is so new). I also see I do not have the option to display alleles in the GO ribbon (which could make sense) and that I need to convert my alleles to genes. Doing this step I get 4 results (3 in *D. melanogaster*) so I’m not sure how it relates to your search in fig 2a but I’m allowed to display a go ribbon once I’ve done this. It might be useful to make this clearer in the manuscript.

It would be nice if the stack viewer wasn’t limited to 100 genes and if it could be compared alongside with other types of data (phenotype, expression...).

The export to Pangea function made things very simple. On the main pangea page “gene annotation subsets” is misspelt (“anotation”). Is the enrichment to Direct GO term only useful to anyone? It might be misleading. Similar on the EXP only? It seems useful to have the option to exclude HTP.

I found the enrichment to slim term option confusing; the results show gene set category IDs – but I’m not sure where I can see what the category IDs correspond to? Wouldn’t it be simpler to display the term name?

Further down in the enrichment table “overlapping” is misspelt “overlapping”.

Line 160: I don’t understand what “different cuts of GO data” means?

Search Genes / Gene Set Category Summary

Copy Excel CSV

Gene Set Category (Category)	Gene Set Category ID	Description	Unique Genes in Category	Species Unique Genes	# Search Genes Found in Category	# Search Genes Not Found In Category	Search Genes Found
GO Biological Processes	42	GO Biological Process Annotations, enrichment uses GO relationships to transit enrichment to parent terms	10832	13968	50	0	FBgn0001995; FBgn00030433; FBgn00031651; FBgn00034001; FBgn00036335; FBgn00037892; FBgn0040907; FBgn0283525
SLIM1 GO BP	11	GO Biological Process, high-level categorization set . 20 terms.	9964	13968	50	0	FBgn0001995; FBgn00030433; FBgn00031651; FBgn00034001; FBgn00036335; FBgn00037892; FBgn0040907; FBgn0283525
SLIM2 GO BP	14	GO Biological Process subset for D.mel, for enrichment analysis . 84 terms.	8789	13968	10	40	FBgn0011787; FBgn0042112; FBgn0283525

Section: Protein and ncRNA structures:

It would be interesting to know if you only display alphafold structures, or do you give the users the option to view/toggle “real” structures where available?

Section: Variant molecular consequences

For the IMPACT glossary, I had to click through to ensemble, and then click through to their glossary. It would be very helpful to have definitions available as a mouse-over tool tip.

Diopt:

I just wanted to say thank you for providing this tool. It is very useful and easy to use!

Quicklinks menu:

I don't know if it is just my browser (firefox 118.0.2 (64-bit)), but the downarrow in the quicklinks menu obscures the item at the bottom, and when pressing down, it goes past the section. Therefore I cannot use the the quicklinks to get to a species of interest, although I can tab with the space bar. The tables are rather big 😊 See screenshots below. It would be nice to be able to middle-wheel scroll, or otherwise manually slowly scroll through the menu instead of this huge jump.

Alignment	Complementat	Transgene ?
		1

- Genomic Location
- Function
- Summaries
- Gene Model and Products
- Expression Data
- Alleles, Insertions, Constructs, and Aberrations
- Variants
- Phenotypes
- Orthologs

Scrolling down one click with the down arrow:

Report Sections ?

Open Close

- Functional
- Complementation
- Interactions
- Pathways
- Genomic Location and Mapping

Valerie Wood
Associate Editor
GENETICS

Nov. 23, 2023
RE: GENETICS-2023-306553

Dear Valerie Wood,

Thank you for accepting our manuscript pending minor revision. We have carefully reviewed the concerns and the suggestions of the reviewers and have revised the manuscript accordingly. Our response is given in a point-by-point manner below. Changes to the manuscript are shown in the 'marked manuscript-GENETICS-2023-306553.pdf' document. Please note that the line numbers in some parts of the revised manuscript might have changed due to the removal of figures. The line numbers referred to below use the new numbering.

Sincerely,
Arzu Öztürk-Çolak
FlyBase Curator

Response to Reviewer #1 Comments

1. Title: Please include "2023" in the title, since there have been, and will be, other updates.

Response: We thank the reviewer for their input. We believe that since any citation of the paper will include the year, having the year on the title would be a bit redundant. Also, there's the complication that the updates described in this manuscript cover the period 2021-2023, while this paper will get formally published in (and thus a year of citation of) 2024! Considering these issues, we feel it is better not to add the year to the title.

2. Line 43: define EBI?

Response: Reviewer's suggestion has been implemented in Lines 42-43.

3. Line 44: I'd say "strains" instead of "stocks".

Response: We've rephrased that part as 'reported mutations in fly genes' in Line 44.

4. Line 45: maybe reword as: "...considering the importance of chemical testing in Drosophila research..."

Response: Reviewer's suggestion has been implemented in Lines 44-45.

5. Line 51 would read more smoothly as: "Since 2021 we have been collaborating with the EMBL..."

Response: Reviewer's suggestion has been implemented in Line 50.

6. Line 47: "New to Flies" should be in quotes.

Response: Reviewer's suggestion has been implemented in Lines 47 and 304.

7. In a few places a ref or two might be helpful, for ex. on line 131.

Response: Reviewer's suggestion has been implemented in Lines 123, 247-251 and 265.

8. Line 304: Please call it: "Outreach resources". Otherwise, one expects to hear how Flybase did outreach, which is not what is meant here.

Response: Reviewer's suggestion has been implemented in Line 296.

9. It might be nice to include brief mention of connections to other model-organism databases, though I realize this is not an update per se. Similarly to human-disease genes (ditto).

Response: As the reviewer surmises, we have not made any significant changes/additions to our links to other MODs or to human disease genes in the last 2 years, so these FlyBase features fall outside the scope of this update paper.

Response to Reviewer #2 Comments

We thank Reviewer #2 for their suggestions on changes to the website. We'll consider them as future improvements but they will take some time to implement.

1. Line 88: Please indicate that the Vocabularies button is located on the main/home/landing page

Response: Reviewer's suggestion has been implemented in Line 91.

2. Line 146-153: could you please elaborate a bit more in this section & provide higher resolution screenshots?

Response: That section has been redrafted to address the reviewer's suggestion. Please see Lines 136-184. The resolution of screenshots has been improved.

3. I assume that fig 2A relates to finding a set of genes with shared function, since it depicts a phenotype search? I searched for the same thing as shown – e.g. affected tissue: Johnston organ I got 41 results – 38 from D. melanogaster – but your search results show '32' (I know this will change with time and curation, but confusing since your manuscript is so new). I also see I do not have the option to display alleles in the GO ribbon (which could make sense) and that I need to convert my alleles to genes. Doing this step I get 4 results (3 in D. melanogaster) so I'm not sure how it relates to your search in fig 2a but I'm allowed to display a go ribbon once I've done this. It might be useful to make this clearer in the manuscript.

Response: We've edited the Figure 2 legend to address the reviewer's suggestion. Below are the steps we used to generate Figure 2B:

- Type in 'Johnston organ' in QuickSearch>Phenotype, click on Search
- On the new Hitlist page (with 41 results), click on Convert and select Genes
- The new Hitlist page should have 32 items.

4. Line 160: I don't understand what "different cuts of GO data" means?"

Response: That section has been redrafted according to the reviewer's suggestion. Please see Lines 154-158.

November 27, 2023

RE: GENETICS-2023-306553R1

Dr. Arzu Öztürk-Çolak
University of Cambridge
Physiology, Development and Neuroscience
Downing Street
Cambridge
United Kingdom

Dear Dr. Öztürk-Çolak:

Congratulations! We are delighted to inform you that your manuscript entitled "FlyBase: updates to the Drosophila Genes and Genomes database" is acceptable for publication in GENETICS. Many thanks for submitting your research to the journal.

To Proceed to Production:

Add oupsupport@scipris.com and genetics.oup@novatechset.com (or the domains @scipris.com and @novatechset.com) to your email program's "safe senders" list. You will be contacted by both at various points during the production process.

1. Format your article according to GENETICS style, as discussed at <https://academic.oup.com/genetics/pages/general-instructions>. Ensure that you comply with data and community resource citation guidelines (<https://academic.oup.com/genetics/pages/general-instructions#Data-Policy>).
2. Upload your final files at <https://genetics.msubmit.net>. The GSA Journals use SciPris to manage article licensing and payment. If you do not have a SciPris account, you will receive an email from no-reply@scipris.com to sign up to use Oxford University Press' author portal. After logging in, follow the online instructions to sign your licence and arrange any payment due.
3. Your currently-accepted manuscript (unedited, as submitted, reviewed, and accepted) will be published at GENETICS and deposited into PubMed as an Advance Access article. Notify sourcefiles@thegsajournals.org before signing your license if you do not wish to publish your article via Advance Access.
4. We invite you to submit an original color figure related to your paper for consideration as cover art. Please email your submission to the editorial office or upload it with your final files. You can submit a small-sized image for evaluation, and if selected, the final image must be a TIFF file 2513px wide by 3263px high (8.375 by 10.875 inches; resolution of 600ppi). Please avoid graphs and small type.

If you have any questions or encounter any problems while uploading your accepted manuscript files, please email the editorial office at sourcefiles@thegsajournals.org.

Sincerely,

Valerie Wood
Associate Editor
GENETICS

Approved by:
Paul Sternberg
Senior Editor
GENETICS

Review comments (if applicable):